# Lung Cancer and Self-Management Interventions: A Systematic Review of Randomised Controlled Trials

**DOI:** 10.3390/ijerph19010536

**Published:** 2022-01-04

**Authors:** Rachel Anne Rowntree, Hassan Hosseinzadeh

**Affiliations:** School of Health and Society, University of Wollongong, Wollongong, NSW 2522, Australia

**Keywords:** lung cancer, lung malignancy, self-management, self-care, home-based

## Abstract

**Background:** Lung cancer is the most common cancer worldwide. Evidence suggests self-management (SM) interventions benefit cancer patients. This review aims to determine the effectiveness of SM interventions for lung cancer patients. **Method:** Searches occurred in PubMed, Cinahl, ProQuest, Psych Info, Scopus, and Medline, using predefined criteria, assessing randomised controlled trials (RCTs). **Results**: Five hundred and eighty-seven studies were yielded, 10 RCTs met criteria. Of the total patient pool, 1001 of 1089 had Non-Small Cell Lung Cancer (NSCLC). Six studies tested home-based SM exercise, two studies SM education, and one each for diary utilisation and symptom reporting. Fatigue was the most targeted function. Other functions targeted included exercise capacity, anxiety, depression, quality of life (QoL), sleep quality, and symptom burden. Six studies met their primary endpoints (five SM exercise, one SM education). Positive outcomes are described for fatigue, anxiety/depression, sleep quality, self-efficacy, and exercise capacity. With exception to fatigue, early-stage NSCLC, younger age, female, never smokers, partnered patients experienced increased treatment effect. **Conclusions**: SM interventions improve outcomes among some lung cancer patients. Interventions targeting fatigue yield benefit despite histology, stage or gender and could encourage broader cohort engagement. Consideration of patient characteristics may predict SM effect. Effectiveness of home-based SM exercise by NSCLC stage and SM tailored to sociodemographic variables requires further research.

## 1. Introduction

Lung cancer is the most common cancer and leading cause of cancer death worldwide, contributing 11.6% of total cancer diagnosis and 18.4% of total cancer deaths [1].

Patients with lung cancer present as highly symptomatic with complex needs [2]. Pain, symptom distress, anxiety, dyspnoea, fatigue, and appetite loss can be found in over 90% of lung cancer patients [3]. Lung cancer treatment adds further complexities to the patient’s symptom burden. Treatment related adverse events for lung cancer include haematological abnormalities, hypertension, pneumonia, and treatment fatigue [4].

Cancer associated costs can further impact patient outcomes [5,6]. Patients managed in both third-party payer and free public health systems experience out of pocket expenses and indirect financial burden such as income loss [7]. The sequalae associated with financial burden for cancer patients sees a reduction in leisure activity, higher dependency on savings and selling of possessions resulting in lowered treatment compliance and higher stress [5,7]. Treatment evolution for lung cancer has shifted this disease to be considered as chronic in several situations [8]. Lung cancer as a chronic disease can exacerbate anxiety, intensify fear of recurrence, hinder life plans, and increase stigma-related challenges [8]. Several guidelines currently exist, focusing on the treatment of lung cancer. Examples include guidelines from the European Society for Medical Oncology (ESMO) and the National Comprehensive Cancer Network (NCCN). These guidelines mainly focus on lung cancer screening, and surgical, radiation and systemic interventions appropriate for treatment of specific histology and stages of lung cancer disease [9,10,11] The “Optimal Care Pathway for People Living with Lung Cancer” endorsed by Cancer Australia, acknowledge the need for supportive care in the management of physical, mental and spiritual associated with lung cancer [12]. None of the mentioned guidelines specifically discuss the role of SM interventions within the lung cancer cohort.

SM intervention among cancer patients is increasing. There has been a shift from the traditional provider to patient relationship. Individuals are increasingly playing a key role in their care [13,14,15]. SM is described as a person’s ability to manage their disease symptoms including treatment, physical, social and lifestyle changes [16,17,18,19]. SM has five identified components that include problem solving, decision making, resource utilisation, patients/provider relationship development and taking action [20,21]. Studies have shown that SM leads to better health, better healthcare, better doctor–patient relationships, and communication, as well as reductions in depression, fatigue, pain, and emergency room visits [22,23,24,25]. In the context of SM and cancer, the National Cancer Research Institute (NCRI) define SM for cancer patients as “*Approaches used by the individual affected by cancer (or life limiting illness) and its effects to optimise living (with the illness and its effects)*” [26].

The main body of evidence for SM as a tool to manage chronic disease has largely been explored in non-malignant disease, although data are emerging that cancer patients utilising SM experience better disease management and improved quality of life [27]. A literature review of SM programmes targeting cancer patients identified six established SM programmes which centred around SM education and SM guided exercise. These programmes were mainly used in patients with breast or prostate cancer due to incidence rates and high survivorship at five years post diagnosis [28]. Nonetheless, SM interventions for cancer patients are lagging other chronic conditions, possibly due to the complex nature of the disease [27]. To the best of our knowledge, there is no other systematic review which assesses the efficacy of SM interventions among those with lung cancer. This systematic review aims to fill this gap by collating all data about the effectiveness of SM interventions on patient outcomes among people living with lung cancer using RCTs.

## 2. Materials and Methods

This systematic review was registered with PROSPERO (registration number CRD42021253619). The Preferred Reporting Items for Systematic Reviews and Meta-Analysis (PRISMA) framework was used for this review. The population, intervention, control, outcome framework forms the primary question “how effective are SM interventions at improving outcomes among patients with lung cancer?” The effectiveness of the SM interventions was determined by the result of the effect measure described in the respective studies. Secondary objectives assessed outcomes by intervention type, stage and histology of disease, patient socio-demographics, and influence of family/partners participation in SM activity. The inclusion and exclusion criteria are described in Table 1.

### 2.1. Search Strategy

Databases searched were EBSCOHOST (CINAHL, Medline, Psych Info, Ipswich, MA, USA) Pro Quest, Scopus, and PubMed. Keywords, Boolean operators, and truncation terms used were “lung cancer” or “lung malignancy” or “thoracic malignancy” or “lung tum*” or “lung aden*” or “lung carcinoma” or “thoracic cancer” or “NSCLC” or “lung neoplasm” and “self manage*” or “self-manage*” or “self car*” or “self-car*” or “self-efficacy*” or “self efficacy*” or “home-base*” or “home base*” or “self-regulat*” or “self regulat*”. Searches occurred in July and August 2021, and keywords commanded to appear in “title” and “title and abstract”. Searches aimed to be identical across all databases to maintain authenticity. No date parameters were enforced to allow for a comprehensive search.

### 2.2. Data Extraction

Endnote (Clarivate Analytics, Philadelphia, PA, USA) and Covidence (Covidence, Melbourne, Australia) software packages were used for collation and extraction purposes. Two reviewers (RR and HH) assessed studies prior to inclusion within this review. The Critical Appraisal Skills Programme (CASP) for RCTs was used to appraise the selected articles. RR conducted the initial appraisals for all studies. Reviewer HH and RR performed final checks on extraction and appraised data. Table 2 outlines the extracted data for each RCT. Table 3 outlines the quality assessment outcomes of the 10 RCTs using CASP for RCTs checklist.

## 3. Results

The primary aim of this review is to establish how effective SM interventions are at improving patient outcomes among people with lung cancer. Due to the variability in the endpoints and their associated effect measures, a narrative synthesis approach has been adopted to describe the findings from the included studies. The heterogenous nature of the interventions and their associated effect measure prohibited the utility of a meta-analysis for this review.

### 3.1. Database Record Yield

Database searches yielded 587 studies; 252 duplications and 201 irrelevant articles were removed. One hundred and thirty-three articles were reviewed for inclusion. Eighty-two percent of studies were the wrong design (*n* = 109), eight RCTs met the inclusion criteria, and a further two studies were identified via reference list pearling. Search findings are documented in the PRISMA algorithm (Figure 1).

### 3.2. Quality Assessment of Included Studies

All studies provided a clear focus of research and were randomly assigned. Seven of the ten studies accounted for all patients at conclusion, one study had failure to follow up in five patients [33] and two studies could not be determined [36,39]. Due to the nature of studies, blinding was not possible for all stakeholders. All groups were treated equally and baseline characteristics, overall, were evenly distributed between control and intervention arms. All studies clearly defined a primary endpoint. Effect measures for endpoints were clearly outlined in all studies with results documented indicating if intervention was significant via a *p* value. Half of the studies specified a 95% confidence interval (CI). All results could be applied to local populations and consideration had been given to the outcomes appropriate for this cohort. One study found the primary endpoint yielded a negative outcome, which could be considered harmful in the intervention arm [34]. Eight of the studies stated that sample size was adequate, and two studies could not be determined. Two of the eight studies stating adequate sample size raised uncertainty of being sufficiently powered to detect an effect within their discussion. Seven of the studies recruited from single centres and three studies recruited over three sites each.

### 3.3. Effectiveness of SM Intervention among Patients with Lung Cancer

The total patient pool was 1089 patients. Baseline targeted function details were recorded for each study and at subsequent time points during the intervention period and at completion. Six of the 10 studies measured effect after the completion of the SM intervention [29,30,31,33,35,36]. Thirty days post intervention was the shortest follow up period [33] and six months from baseline was the longest follow up period [29,30,31]. The SM education studies were executed in the hospital setting [35,36] with Schofield et al. allowing for home-based education in the event a patient was not well enough to attend clinic. The remaining eight studies were conducted in the participant’s home or community. Six of the 10 studies met their primary endpoint which, on balance, supports the hypothesis that SM interventions utilised among patients with lung cancer are effective.

### 3.4. Outcomes by SM Intervention Type

Four intervention types were identified, targeting eight different patient functions. Exercise was the intervention of choice in 60% of the studies. Two studies utilised SM education [35,36], one study used a telephone-based symptom reporting tool [37] and the remaining study adopted a QoL diary [34]. Five of the six studies adopting exercise as the intervention met their primary endpoint. Telephone symptom reporting and a QoL diary failed to meet their primary endpoints. SM education provided mixed results, with one study meeting their primary endpoint.

Fatigue was the targeted function in three studies, all meeting their primary endpoints. One of the studies primarily targeting fatigue also had a primary endpoint for self-efficacy which also demonstrated significance [39]. Two studies focused on exercise capacity, with one study meeting its endpoint [33] and the other did not show significance [31]. The remaining five studies focused on various functions, with differing primary endpoint outcomes. Chen [29,30] demonstrated positive outcomes for anxiety/depression and sleep quality. The studies by Chen et al., were conducted in the same patient sample. In contrast, studies focusing on QoL, unmet needs, and symptom burden did not meet their primary endpoint [34,35,37]. Table 4 outlines these findings.

### 3.5. Outcomes for Studies including and Excluding Early-Stage NSCLC

Lung cancer histology was available for 1077 of the 1089 patients. NSCLC was reported for 92% of those 1077 patients (*n* = 1001). Staging for the 1001 NSCLC patients was documented in 8 of the 10 studies as stage I-IV, although one study amalgamated stage of disease for SCLC and NSCLC participants [38]. Definitive staging of NSCLC was available for 702 of the 1001 pooled NSCLC cohort (70%). Stage I-II accounted for 41% of all definitively staged NSCLC disease (*n* = 294) 25.5% was stage III NSCLC disease (*n* = 182) and 33% was stage IV NSCLC disease (*n* = 235). 

Six of the 8 NSCLC staged studies indicated the inclusion of early-stage NSCLC disease (stage I-II), with five of the six studies availing the exact numerical breakdown by NSCLC stage. Of the 462 participants included within the 5 trials demonstrating a clear breakdown by NSCLC stage, most patients (63.6% *n* = 294) presented with early-stage disease. Only one of these five studies reported no significant difference between the control and intervention group for the primary endpoint [31]. This study accounted for only 1% (*n* = 3) of early-stage disease, with most participants having advanced disease. The four studies that account for the remaining 291 early-stage patients demonstrated significance across all primary endpoints, which included benefits to subjective and objective sleep [30], anxiety and depression [29], perioperative functional capacity [33] and fatigue [39]. The study where staging for SCLC and NSCLC was amalgamated [38], NSCLC accounted for most of the patient population (82% *n* = 75) and the study met its primary endpoint. All six studies including early-stage disease utilised physical activity as the primary intervention (with supplementary education and diary utilisation provided in five of these studies) to influence the effect measure. On balance, 83% of studies including early-stage disease met their primary endpoint.

Four studies excluded early-stage NSCLC, or their inclusion could not be determined [34,35,36,37], two of the studies staged their NSCLC participants (stage III-IV). The remaining studies stated “inoperable” NSCLC, which is more common in later stage (IIIb-c/IV) disease [40]. Three of these four studies included patients with SCLC and two for mesothelioma. Nonetheless, 78% of patients had NSCLC (*n* = 419). In contrast to interventions explored in studies including early-stage disease, interventions were more heterogenous in this group. Interventions included SM education [35,36], self-reporting of symptoms [37], and keeping a diary [34]. Outcomes for studies excluding early disease demonstrated mixed results. Only one of these four studies demonstrated benefit to the intervention group [36], this study was exclusively stage III-IV NSCLC patients and targeting fatigue. The remaining three studies which included SCLC, mesothelioma, and advanced NSCLC, failed to meet their primary endpoint. These three studies included patients with lower performance status when compared to Wangnum et al. and did not target fatigue. The study assessing the utility of keeping a diary demonstrated a decline in QoL in the intervention group suggesting the intervention had a negative effect on patient outcomes [34].

### 3.6. Outcomes for SCLC and Mesothelioma

SCLC and mesothelioma accounted for 6.2% and <1% of confirmed histology patients, respectively (*n* = 68 and *n* = 8). SCLC was represented in four of the 10 studies, with only one of those studies meeting the primary endpoint, which targeted fatigue [38]. The two studies including mesothelioma did not reach their endpoint [34,35]. The representation of SCLC and mesothelioma was small in comparison to NSCLC (92% versus 7%) therefore a degree of caution should be used in applying the findings to these cohorts.

### 3.7. Outcomes by Operability Status

Five of the 10 studies stipulated operability within their inclusion criteria. Three studies listing inoperable lung cancer as part of their inclusion criteria derived no benefit in the intervention group [31,34,35]. In contrast, studies where operable lung cancer made up part of the inclusion criteria, described a benefit in the intervention arm [33,39]. The latter two studies were restricted to patients with stages I-III NSCLC.

Overall, studies including early-stage NSCLC (I-II) and/or operable disease were more likely to meet their primary endpoint, which contrasted with studies excluding early-stage disease, including SCLC and mesothelioma and inoperable disease.

### 3.8. Patient Socio-Demographics and SM Outcomes

Socio-demographic variables describe the characteristics associated with individuals. Age, sex, education, smoking status, relationship/living arrangements have been identified as variables that can influence health outcomes [41,42,43] Background details for each variable is documented in Appendix A. Table 5 tabulates sociodemographic variables for each study included in this review.

### 3.9. Age

Participant age was recorded for all trials, with eight of the 10 studies recording a mean age for control and intervention groups. The pooled mean age for these eight studies is 61.86 years, which is almost one decade younger than the median age of lung cancer presentation [40]. The remaining two studies categorised age as ≤60 y/61–70 y/≥70 y [34] and ≤60 y/≥60 y [38]. Mills et al. had 32.5% of participants aged over 70 y. In contrast, age 70 y and above could not be determined in Zhang et al., although most patients (56%) were 60 years or younger. One study excluded patients over the age of 65 y [36] and another excluded patients over 70 y [33]. Both studies recorded positive outcomes in the intervention arms. In contrast, Mills et al., with one third of patients over 70 years did not demonstrate a positive outcome with the intervention. Of the six trials which showed a significant outcome, one excluded people over 65 years, one excluded over 70 years and one study included most participants less than 60 years. This review highlights two findings in reference to age. The first is that participants, overall, are younger than the average median age of lung cancer presentation which supports the argument of study outcomes being difficult to extrapolate to older cohorts. Secondly, where significance was reached, three studies were weighted to patients under 60 years. The latter suggests that younger patients may derive a larger benefit than their older counterparts.

### 3.10. Sex

Participant gender was evenly split when studies were pooled together. Female participants equated to 50.1% (*n* = 546). Of the six studies demonstrating significance, four contained more females, with a mean percentage of 59% [29,30,39]. This suggests that females have a larger benefit when utilising SM interventions compared to males. The remaining two studies with significance [36,38] weighted more to male participants. The latter studies were largely represented by participants under 60 years of age. This raises the question of younger age mitigating any adverse association with being male, having lung cancer and utilising SM interventions?

### 3.11. Education

Seven of the 10 studies documented participants’ level of education attainment. Five of these studies had significant outcomes, although the level of education was mixed and makes any association with education attainment and SM outcomes inconclusive. Further, the additional two studies that have not shown significance, most participants had completed high school or higher, thus further supporting an inconclusive finding.

### 3.12. Smoking Status

Three of the included studies recorded smoking status for participants. Two studies demonstrated significance and most patients were never smokers [33,38]. In contrast, Eadebrooke et al., contained a majority of current and former smokers and did not meet its endpoint.

### 3.13. Relationship Status and Living Arrangements

Half of the RCTs documented the martial and/or living arrangements for participants. All five studies documented most patients were either married and/or not living alone. Eighty percent of these studies (*n* = 4) met their primary endpoint.

### 3.14. SM Interventions Involving Family and Caregivers

None of the included studies focused on the involvement of family caregivers participating in SM interventions for patients with lung cancer. The inclusion criteria specified that interventions needed to be tested against a standard care control arm. Database searches yielded two RCTs that included family caregivers in the SM of patients with lung cancer. Both studies were excluded due to the wrong comparator being used. The studies were comparing the effectiveness of two interventions and did not include a standard care arm [44,45]. Both studies document a benefit to the involvement of family caregivers and SM interventions among patients with lung cancer, although the lack of a standard care arm makes it difficult to measure the degree of effect. This was acknowledged by Porter et al. [45].

## 4. Discussion

This review aims to assess the effectiveness of SM interventions among people with lung cancer. Overall, the utility of SM within this cohort appears to yield a positive effect. From the 587 database results, 453 results were removed (252 duplications and 201 irrelevant articles) of the remaining 133 screened articles, and pearling of references, 10 RCTs were included within this review. Overall, the majority of the included RCTs demonstrated that the intervention of SM among lung cancer patients exhibited a positive effect on patient outcomes and therefore, at first glance, favours the hypothesis that SM interventions are beneficial among patients with lung cancer. The studies however, tested four intervention types, targeting eight different functions, and included patients with three different histology, various disease staging, operability, and different sociodemographic variables. The heterogenous nature of the review makes it difficult to affiliate the hypothesis as a blanket rule for all lung cancer patients and a more tailored consideration is required.

All included articles provide sound rationale for their study, with clear commentary of their research aims and primary endpoints. Results were provided in conjunction with a *p* value. Further, all studies have clear randomisation and overall, equal distribution of characteristics are seen between the intervention and control arms. Eight of the studies specified that the sample size was adequate to detect an effect, although two of these studies commented on their uncertainty of being sufficiently powered [31,35]. The remaining two studies could not be determined [38,39]. When a study is not adequately powered, it can prohibit the research question from being answered as accurately as possible, which may be the case for up to four of the studies included within this review [46].

Seven of the studies were performed from single centres, which raises the question of result validity. Interestingly, all multiple site trials did not meet their primary endpoint and six of the seven single centres did meet their endpoints. Half of the studies had less than 100 participants. Small sample sizes increase the risk of distorting the results [47]. Single centre and small sample size were common themes identified as limitations throughout this review.

Due to the nature of the intervention, none of the studies were fully blinded, which introduces an element of bias to this review. A systematic review of RCTs that used patient reported outcomes which included blinded and non-blinded patients found that the effect size was exaggerated among unblinded patients [48].

Physical exercise has been proven in other studies covering various malignancies to reduce fatigue, depression, improve sleep, and improve clinical/functional outcomes which is documented in other systematic reviews with meta-analyses [49,50]. This review builds upon existing evidence that exercise yields improved patient outcomes among people with lung cancer. All 6 studies utilising SM exercise included early-stage NSCLC patients, and this cohort accounted for most of the pooled patients. There is existing data that exercise in early-stage lung cancer has positive effects [51]. This review however supports home-based exercise in this cohort which could be considered a cost-effective alternative to hospital-based programmes. Advanced NSCLC representation was not seen in all SM exercise studies. This makes it difficult to say with confidence if the effect of SM exercise was equally effective in early and later stage NSCLC. Compared to early stage disease, evidence suggests exercise outcomes are less certain in later stage disease, which should be considered in light of this review [51]. A systematic review of RCTs for hospital-based exercise in advanced lung cancer demonstrated a significant improvement for exercise capacity and QoL, but no improvement for fatigue, depression, and anxiety [52]. This finding contrasts to the current review for advanced lung cancer patients. The intervention setting was different between the two reviews (hospital based versus home-based) and the current review includes early-stage patients, which may have influenced the outcomes.

Beyond early-stage disease, SM exercise trials that were weighted to the female gender and evidence of being partnered demonstrated improved outcomes. Evidence states that there are some genetic and hormonal factors that influence cancer between the sexes [53]. Females experience better survival when surgery, radiotherapy and chemotherapy modalities are utilised when compared to men [53]. Marriage appears to have a desirable effect on cancer mortality rates. Further, social deprivation might be a factor leading to more advanced disease, co-morbidity, treatment morbidity and treatment access [54] It could be argued that the benefits of female gender and partnership yield a larger effect from exercise intervention.

In contrast to SM exercise studies, non-exercise studies tested a range of interventions across patients with more advanced and/or inoperable disease and various histology. The heterogeneity in interventions makes it difficult to draw a definitive conclusion. Only one of the non-exercise-based SM interventions met its primary endpoint (fatigue), which further supports that SM interventions are more effective in earlier stage NSCLC. Further, non-exercise interventions contained more males which supports the finding that SM interventions work better in females.

Early stage lung cancer is diagnosed in approximately 40% of patients, with locally advanced and advanced stage equating to 20% and 40%, respectively [40]. Goals of treatment are often different among these cohorts, ranging from a surgical and curative intent approach among stage I-IIIa NSCLC, to a palliative approach in the advanced setting [40]. SCLC, which accounts for approximately 13% of all lung cancers, and mesothelioma (incidence varies globally) both have poor prognosis with the aim of treatment being palliative care [40]. Patients with lung cancer often present with several co-morbidities which may influence function [55] It could be argued that different disease presentation and associated treatment pathways may influence how effective SM interventions among those with lung cancer, and possibly warrants further research.

Smoking is the biggest risk factor for developing lung cancer [56] and this may explain why only three of the trials included smoking status within their patients characteristics. Not surprisingly, studies represented by majority non-smokers met their endpoint which contrasted to the study containing majority smokers. Evidence for patients developing lung cancer and have a smoking history experience more guilt, shame, and perceived stigma than their non-smoking counterparts [57], further, evidence shows that ever smokers have a poorer performance status when compared to never smokers [58]. Smoking history is scant within this review. The trend seen for effective SM outcomes in non-smokers, however, raises the need to develop SM interventions that mitigates any prohibitive effect of being a current or former smoker.

The mean age could be determined for 80% of the pooled patients and shown to be almost a decade lower than the median age at diagnosis for lung cancer; consideration should be given to the extrapolation of this review to elderly patients. This limitation is not exclusive to this review, studies world over tend to recruit subjects that are aged 18 to 65 years, despite the elderly being one of the fastest growing populations worldwide. Fortunately, the sea of change is approaching with medical agencies in Europe, Canada and India making recommendations for adequate geriatric representation in medical trials moving forward [59].

Half of the studies recorded if patients were partnered or lived alone, with most participants identifying as married or not living alone. This review supports evidence that there are improved outcomes in cancer among those with a partner [60]. This review failed to test if partner/family involvement with SM intervention improved outcomes. Two studies identified in the search included family participation when utilising SM interventions. The studies however compared the effect of two interventions and no comparison was made against a control arm, which excluded them from this review. Family involvement of SM interventions tested against a control arm warrants further research to establish the true effect of their role in SM interventions for people with lung cancer.

Irrespective of histology, disease stage, and some sociodemographic presentation, fatigue as a primary endpoint was significantly improved in all three studies included in this review. Fatigue is the most frequently reported adverse event associated with lung cancer and is seen from diagnosis until end of life in 57–100% of lung cancer patients. Fatigue experienced by patients with lung cancer can have a negative impact on QoL [55]. This finding could help incentivise patients to adopt SM interventions as a proven way to mitigate the effects of fatigue and extends the benefit of SM beyond those with early-stage NSCLC disease, female gender, never smokers and partnered.

This systematic review has several limitations. All studies were not fully blinded, which may lead to bias. Small sample size and single centre trials were evident in 50% and 70%, respectively, and may distort the validity of the outcomes. None of the studies had follow up beyond six months which limits the durability of the SM interventions beyond this time frame. Two of the studies highlighted the low-cost low-technology natures of their trials, which may have introduced administrative errors. Two studies reported difficulty in patient recruitment and high attrition rates suggesting the engagement with this cohort is challenging. Three studies highlighted the challenges in overseeing activities in the control arm of their respective studies and one study highlighted that relying on patient adherence to the intervention and any subsequent patient reporting was difficult. These confounding factors may have influenced the results within this review.

## 5. Conclusions

This review demonstrates that the role of SM in lung cancer appears to have a positive effect on improving patient outcomes. Exercise was the most used intervention and was largely effective in at improving features of fatigue, sleep quality, anxiety, depression, and exercise capacity. The effect of SM interventions appears to be most pronounced in early-stage NSCLC disease. Being female, partnered and a non-smoker also appear to support improved SM intervention outcomes. Fatigue yielded positive outcomes irrespective of patient features and interventions, demonstrating utility across the broad spectrum of this disease, which is clinically meaningful. More research is needed to determine effectiveness of home-based SM exercise by NSCLC disease stage. Further, this review identifies that the development of tailored SM interventions which consider disease stage/histology, gender, partner status, smoking history and age should be explored to enhance a more precise SM intervention effect for all lung cancer patients.

## Figures and Tables

**Figure 1 ijerph-19-00536-f001:**
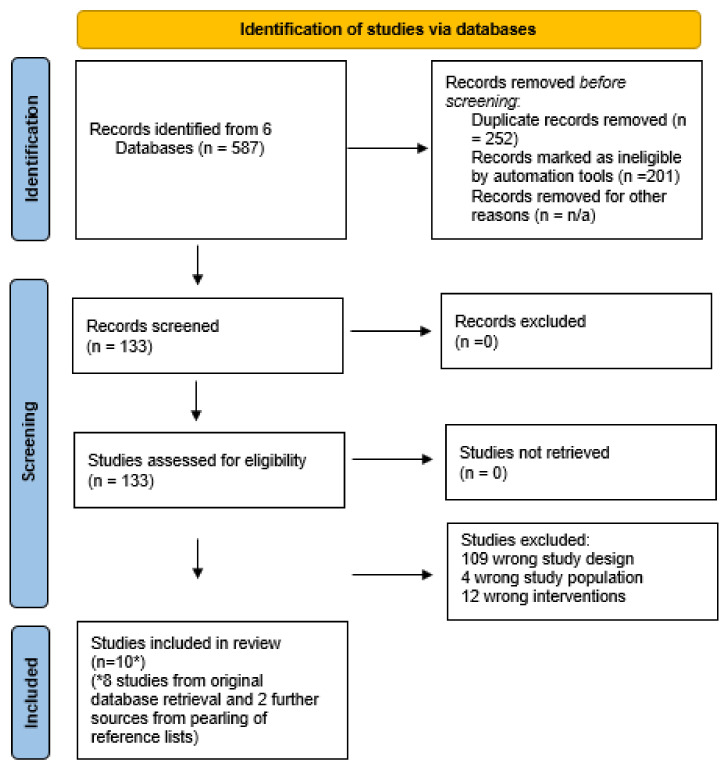
Prisma Flow Chart. * confirms that two further studies were yielded via database pearling.

**Table 1 ijerph-19-00536-t001:** Study Criteria.

Characteristics	Inclusion	Exclusion
Patients	All stages and histology of lung cancer Adults 18+ years No restriction for − Demographics − Gender − Socioeconomic status patient involvement +/− family or caregivers in SM intervention	Mixed cancer cohort studies where lung cancer cannot be analysed exclusively Non lung cancer patients
Intervention	SM interventions that patients +/− their family caregivers actively participated in compared to standard care as determined by study. The intervention is hypothesised to improve patient outcomes associated with lung cancer	Studies lacking a control arm which utilises standard or usual care Studies not assessing efficacy of intervention
Outcomes	Change in effect measure associated with the SM intervention	
Study Design	RCTs—measuring efficacy of intervention against standard care, published, peer reviewed, English language, and full text available.	Pilot RCTs RCTs assessing for feasibility All other trial designs. All secondary data All duplications Non English language studies

**Table 2 ijerph-19-00536-t002:** Extraction Details of Included RCTs.

Author, Year, Study Country	Sample Size in Both Control and Intervention Group	Age and Sex in Control and Intervention Groups	Stage and Histology of Disease in Control and Intervention Groups	Study Setting	Intervention Type	Intervention Delivery Method	Primary and Additional Endpoints
[29] Taiwan	Intervention (*n* = 58) Control (*n* = 58)	Mean age of intervention group 64.76 years with 26 males and 32 females Mean age of control group 63.57 years with 28 males and 30 females	Histology not confirmed, but staging suggests Non-Small Cell Lung Cancer (NSCLC) Intervention Stage 1 (*n* = 34) Stage 2 (*n* = 5) Stage 3 (*n* = 6) Stage 4 (*n* = 5) Stage unknown (*n* = 8) Control Stage 1 (*n* = 41) Stage 2 (*n* = 4) Stage 3 (*n* = 5) Stage 4 (*n* = 4) Stage unknown (*n* = 4)	home	Exercise, (supplementary counselling, and diary utilisation.)	Moderate intensity walking exercise programme consisting of 40 min sessions 3 times per week and weekly exercise counselling. An exercise booklet was given to participants to record their exercise experiences. The programme ran for 12 weeks. Effect measures were recorded at baseline, with follow up (f/u) at 3 months and 6 months. The control group received the same care as the intervention group except the home-based walking programme and weekly exercise counselling. The control group were asked to maintain usual activity and not perform additional exercise within the study period	Primary endpoints are anxiety and depression. Secondary endpoints are severity of cancer symptoms (pain, fatigue, nausea, sleep disturbance, sadness, shortness of breath, difficulty remembering, poor appetite, drowsiness, dry mouth, distress, vomiting and numbness)
[30] Taiwan (a sub-study of Chen et al., 2015)	Intervention (*n* = 56) Control (*n* = 55)	Mean age of intervention 64.64 years with 24 males and 32 females Mean age of control 62.51 years with 25 males and 30 females	Histology not confirmed, staging suggests NSCLC Intervention Stage 1 (*n* = 34) Stage 2 (*n* = 5) Stage 3 (*n* = 5) Stage 4 (*n* = 5) Stage Unknown (*n* = 7) Control Stage 1 (*n* = 38) Stage 2 (*n* = 4) Stage 3 (*n* = 5) Stage 4 (*n* = 4) Stage Unknown (*n* = 4)	home	Exercise, (supplementary diary utilisation.)	Moderate intensity walking exercise programme consisting of 40 min sessions 3 times per week and weekly exercise counselling. An exercise booklet was given to participants to record their exercise experiences. A sleep diary was used to record bed and wake times. The programme ran for 12 weeks. Effect measures were recorded at baseline with f/u at 3 months and 6 months. The control group received the same care as the intervention group except the home-based walking programme and weekly exercise counselling. The control group were asked to maintain usual activity and not perform additional exercise within the study period	Primary endpoints were improvement in subjective and objective sleep quality, and stabilising rest-activity rhythms. Secondary endpoint moderating effect of rest activity rhythms on subjective and objective sleep quality.
[31,32] Australia	Intervention (*n* = 45) Control (*n* = 47)	Mean age of intervention 64.6 years with 22 males and 23 females Mean age of control 62.5 years with 29 males and 18 females	Intervention NSCLC Squamous (*n* = 11) Adenocarcinoma (*n* = 32) Large Cell/other (*n* = 2) Stage IA-IIB (*n* = 2) Stage IIIA (*n* = 11) Stage IIIB (*n* = 6) Stage IV (*n* = 22) Recurrent (*n* = 4) Control NSCLC Squamous (*n* = 10) Adenocarcinoma (*n* = 32) Large Cell/other (*n* = 5) Stage IA-IIB (*n* = 1) Stage IIIA (*n* = 13) Stage IIIB (*n* = 5) Stage IV (*n* = 26) Recurrent (*n* = 2)	home	Aerobic and resistance exercise with supplementary diary utilisation.	8 weeks of individually tailored aerobic exercise (walking, swimming, or cycling) and resistance training. Aerobic exercise starts at minimum of 10 min per session twice weekly up to 150 min per week at study cessation. Resistance exercises included squats, sit-to-stand, heel raises, step-ups, unilateral shoulder elevation, wall press and unilateral shoulder horizontal extension. performed 10 repetitions of each and aim for 80% of all resistance exercise. Hand weights, FitBit Zip©, smart phone supplied. Effect measures were recorded at baseline with f/u at 9 weeks and 6 months. The control arm received the usual care as per hospital protocol and did not receive any exercise advice, or physiology/exercise physiology consultation. The control arm received monthly welfare calls but were not provided with exercise or symptom advice	The primary endpoint is change in functional exercise capacity measured via 6 min walking distance. Secondary outcomes are Physical Activity and Health related Quality of Life (HRQoL)
[33] China	Intervention (*n* = 37) Control (*n* = 36)	Mean age of intervention 56.2 years with 12 males and 25 females Mean age of control 56.2 years with 11 males and 25 females	Intervention NSCLC stage I-II (*n* = 33) Stage III (*n* = 4) Control NSCLC stage I-II (*n* = 32) Stage III (*n* = 4)	home	Aerobic and resistance Exercise, (supplementary nutrition and relaxation education, diary utilisation)	2 weeks of aerobic (3 ×p/w 30 min) and resistance training (2 × p/w 4 actions 10–12 repetitions). Protein whey supplementation provided. Three-day food recall diary. Imagery visualisation with music relaxation prior to sleeping. Music player supplied. Effect measures recorded at baseline, 1 day before surgery and 30 days post-operatively. The control group received usual clinical care which included anaesthetic assessment, drug recommendation for chronic conditions, and smoking cessation and abstinence. No specific recommendations were given for diet, exercise, or mental health.	Primary outcome was perioperative functional capacity (via 6 min walking test). Secondary outcomes included pulmonary function, disability and psychometric evaluations assessed perioperatively.
[34] UK	Intervention (*n* = 57) Control (*n* = 58)	Intervention <60 years (*n* = 21) 61–70 years (*n* = 18) 70+ years (*n* = 18) Males (*n* = 34) Females (*n* = 23) Control <60 years (*n* = 20) 61–70 years (*n* = 18) 70+ years (*n* = 20) Males (*n* = 35) Females (*n* = 23)	Intervention Inoperable stage disease NSCLC (*n* = 45) Small Cell Lung Cancer (SCLC) (*n* = 10) Mesothelioma (*n* = 2) Unknown primary (*n* = 0) Control Inoperable stage disease NSCLC (*n* = 48) Small Cell Lung Cancer (SCLC) (*n* = 9) Mesothelioma (*n* = 0) Unknown Primary (*n* = 1)	home	Diary utilisation	Intervention involved weekly completion of a QoL questionnaire in a diary format. Patients were encouraged to share their diaries with their health care team. Effect measures were recorded at baseline with f/u at months 2 and 4. The control group received standard care. No further details are described within the study	Primary endpoint is QoL. Secondary endpoints were other indices of QoL, diary utilisation, communication, discussion of problems and satisfaction with care
[35] Australia	Intervention (*n* = 55) Control (*n* = 53)	Mean age of intervention 62.3 years with 31 males and 24 females Mean age of control 63.8 years with 34 males and 19 females	Control. SCLC (*n* = 5) NSCLC (*n* = 45) Mesothelioma (*n* = 3) Intervention SCLC (*n* = 4) NSCLC (*n* = 48) Mesothelioma (*n* = 3)	Clinic or home	education	2 × telephone consultations Consult one: individual needs, symptom management, practical support, psychological therapy, and spiritual support Consult two: reinforced important information and self-care advice. Six self-care modules were given to patients for reading at home. Effect measure recorded at baseline, 8 weeks post treatment and 12 weeks post treatment. The control arm received care as advised by the hospital protocol. This involved consultation with a nurse and referral to allied health services if necessary	Primary endpoint reduction in unmet needs, Secondary endpoints: psychological morbidity, distress, and HRQoL.
[36] Thailand	Intervention (*n* = 30) Control (*n* = 30)	Age range for study 45–65 years Mean age of intervention 54.83 years with 19 males and 11 females Mean age of control 57.37 years with 22 males and 8 females	Stage III-IV NSCLC Control Stage III (*n* = 13) Stage IV (*n* = 17) Intervention Stage III (*n*= 12) Stage IV (*n*= 18)	clinic	education	4 × self-care education sessions provided by the consultant, nutritionist (what to eat whilst on chemotherapy) physical therapist (breathing and physical exercise in the home setting) and a psychological nurse (looking after yourself on chemotherapy) 15 min of pre-reading at home prior to consults. This was a nine-week education programme. Effect measures were recorded at baseline and after the study period (exact time point not disclosed). The control arm received education for 30 min from a nurse only on exercise whilst on chemotherapy	Primary endpoint fatigue. Secondary endpoints depression, nutritional status, weight, albumin levels, physical fitness,
[37] USA	Intervention (*n* = 123) Control (*n* = 130)	Mean age of intervention 61 years with 57 males and 66 females Mean age of control 60.2 years with 68 males and 62 females	NSCLC stage IIIa, IIIb, IV, SCLC Control Stage IIIa (*n* = 15) Stage IIIb (*n* = 25) Stage IV (*n* = 70) SCLC (*n* = 17) Intervention Stage IIIa (*n* = 16) Stage IIIb (*n* = 28) Stage IV (*n* = 64) SCLC (*n* = 15) insufficient information for 11 patients	home	Telephone symptom reporting	Weekly symptom reporting by patients via the telephone using a technology-based telecommunication system called SyMon-L for 12 weeks. Effect measures recorded at baseline then f/u occurred at 3,6,9, and 12 weeks. The control arm only monitored symptoms. The delivery of significantly reported symptoms was automated to the care team for further assessment	Primary endpoint symptom burden over 12 weeks. Secondary endpoints the benefit to HrQoL, treatment satisfaction, perceived barriers to symptom management, self-efficacy.
[38] China	Intervention (*n* = 47) Control (*n* = 44)	Intervention <60 years (*n* = 25) 60 years + (*n* = 22) males (*n* = 37) females (*n* = 10) Control <60 years (*n* = 26) 60 years + (*n* = 18) males (*n* = 31) females (*n* = 13)	SCLC and stage I-IV NSCLC Control SCLC (*n* = 7) NSCLC (*n* = 37) Stage I (*n* = 1) Stage II (*n*= 4) Stage III (*n* = 11) Stage IV (*n* = 28) Intervention SCLC (*n* = 9) NSCLC (*n* = 38) Stage I (*n* = 2) Stage II (*n* = 4) Stage III (*n* = 10) Stage IV (*n* = 31)	Home or community setting	exercise	12-week programme of eight forms of simplified Yang style Tai-Chi exercise, performed on day 10 of 21 of 4 courses of chemotherapy, between 8 am–10 am starting with 5–10 min warm up. Taught by instructor or to follow instructional DVD. Effect measures recorded at baseline with f/u at 6 and 12 weeks. The control group performed low impact exercise including arm, neck and leg circles, upper and lower body stretches, and deep abdominal breathing. The control group followed the same timeframe as the interventional group	Primary endpoint, Cancer Related Fatigue (CRF); change in total score of the Multidimensional Fatigue Symptom Inventory Short Form (MFSI-SF). Secondary endpoints, changes in the five subscales scores of the MFSI-SF.
[39] China	Intervention (*n* = 35) Control (*n*= 35)	Mean age of intervention 67.95 years with 13 males and 22 females Mean age of control 67.21 years with 15 males and 20 females	Operable stage I-III NSCLC Control Stage I (*n* = 22) Stage II (*n* = 8) Stage III (*n* = 5) Intervention Stage I (*n* = 21) Stage II (*n* = 10) Stage III (*n* = 4)	home	Exercise (supplementary diary utilisation)	Six weeks of walking exercise. With exercise activity recorded in a diary. Week one included patients walking for 5 min per day, 5 days per week. Self-efficacy assessed weekly. If score >70%, daily walking time increased by 5 min. Effect measures were recorded at baseline, 3 days post-operative, then at weeks 1, 2, 3, 4, 5, and 6. The control group utilised a conventional rehabilitation intervention with telephone f/u conducted in line with the intervention group	Impact of intervention on cancer related fatigue severity and self-management efficacy

**Table 3 ijerph-19-00536-t003:** CASP Checklist for RCTs.

	Checklist Questions	Trial Address Clearly Focused Issue	Assignment of Patients Randomised	Participants Entered to Trial Accounted for at Conclusion	All Stakeholders Blinded to Treatment	Groups Similar at Start of Trial	Groups Treated Equally	How Large Was Treatment Effect?	How Precise Was the Estimate Effect	Results Applicable to Local Population	Were All Clinically Important Outcomes Considered	Are the Benefits Worth the Harms and Cost
Study												
[29]		yes	yes	yes	no	yes	yes	Primary endpoint defined, effect measure Hospital and Anxiety Depression Scale (HADS) Anxiety and depression scores significantly better at 3+ 6 months for both (*p* = 0.0009 and 0.006 anxiety) (*p* = 0.00006 and 0.004 depression). Study sufficiently powered.	95% confidence interval (CI)	yes	No (sub-study Chen et al., 2016)	yes
[30]		yes	yes	yes	no	yes	yes	Primary endpoints defined. Effect measures subjective sleep quality Pittsburgh sleep quality index (PSQI) significantly better in intervention (*p* = 0.001) Objective sleep measure total sleep time (TST) (*p* = 0.023). Other measures for objective sleep were not significant. Study was sufficiently powered	CI not specified	yes	yes	yes
[31,32]		yes	yes	yes	no	15/62 patient characteristics had a >20% difference at baseline	yes	Primary endpoint defined. Effect measure of 6 min walking distance (6MWD) at baseline, 9 weeks (*p* = 0.308), and 6 months (*p* = 0.979) results not significant. Study theoretically powered although authors unsure if this was the case.	95% CI	yes	yes	yes
[33]		yes	yes	5 lost to f/u	no/single blind	yes	yes	Primary endpoint defined. Effect measure Perioperative Functional Capacity, measured as 6MWD 1 day before and 30 days post-operative. 6MWD 60.9 m higher in intervention (*p* < 0.001) other endpoints not significant. Study sufficiently powered (70 min)	95% CI	yes	yes	yes
[34]		yes	yes	yes	no	yes	yes	Primary endpoint defined. QoL Effect measure Trial Outcome Index not significant (*p* = 0.10 and *p* = 0.07) Powered to detect a change and accommodate a 20% attrition rate.	95% CI	yes	no	no
[35]		yes	yes	no	no	yes	yes	Primary endpoint defined. Effect measures were Needs Assessment for Advanced Lung Cancer Patients, HADS, Distress Thermometer (DT) and QoL questionnaire. Study did not recruit enough patients to detect a small effect of the primary outcome. Theoretically adequate sample but not likely to be sufficient. None of the measures were significant (all *p* > 0.10)	95%CI	yes	yes	yes
[36]		yes	yes	cannot tell	no	yes	yes	Primary endpoint defined. Effect measure was Fatigue Assessment tool created by Piper et al. 1998. Significantly better in intervention (*p* = 0.036) Recruitment was adequate for sample analysis	CI not specified	yes	yes	yes
[37]		yes	yes	yes	no	yes	yes	Primary Endpoint defined. Effect measure was Symptom Distress Scale (SDS). Study powered sufficiently to detect a difference between the control and study arm. No significant difference between groups (*p* = 0.505)	CI not specified	yes	yes	yes
[38]		yes	yes	yes	no	yes	yes	Primary endpoint defined. Effect measure change in Multidimensional Fatigue Symptom Inventory—Short Form (MFSI-SF). Intervention significantly better than control (*p* < 0.05). Cannot tell if sufficiently powered	CI not specified	yes	yes	yes
[39]		yes	yes	cannot tell	no	yes	yes	Two primary endpoints defined. Effect measure for fatigue cancer-related fatigue severity, and brief fatigue score. Self-management efficacy effect measure 30 min continuous walking efficacy. All endpoints significantly better in experimental arm at 6 weeks (both *p* < 0.05). Cannot tell if sufficiently powered.	CI not specified	yes	yes	yes

**Table 4 ijerph-19-00536-t004:** Function and Intervention Outcomes.

Study/Study Features	Primary Function Targeted and Effect Measure	Intervention: Exercise	Intervention: Education	Intervention: Telephone Symptom Monitoring	Intervention: QoL Diary	Result of Primary Endpoint Final Follow Up
[29]	anxiety/depression (measured by Hospital Anxiety and Depression Scale (HADS))	✓				Changes in anxiety scores at 6 months −2.18 intervention v 0.79 control *p* = 0.006 Changes in depression score at 6 months −2.57 intervention v 0.88 *p =* 0.004
[30]	subjective sleep (measured by Pittsburgh Sleep Quality Index, PSQI)/objective sleep quality (1° measure by total sleep time, TST)	✓				The PSQI (Wald w2 ¼ 15.16, *p* ¼ *p =* 0.001) TST (Wald w2 ¼ 7.59, *p* ¼ *p =* 0.023)
[31]	exercise capacity (measured assessed by change in 6 min walking distance (6MWD)	✓				The ITT analyses involving all 92 participants for the 6MWD revealed no significant between-group differences. Mean difference (95% CI) 41.34 (−26.67 to 109.35) *p* = 0.232
[33]	exercise capacity (measured by change in six minute walking test (6MWT))	✓				The average 6MWD was 60.9 m higher perioperatively in the prehabilitation group compared to the control group (95% confidence interval [CI], 32.4–89.5; *p* < 0.001)
[34]	quality of life (measured by Trial Outcome Index (TOI) a subset of the Functional Assessment of Cancer Therapy–Lung (FACT-L))				✓	no evidence of a difference in TOI, the primary outcome measure, between the two groups; 95%CI *p* = 0.1
[35]	reduction unmet needs (multiple measures used for assessment)		✓			None of the primary contrasts of interest were significant (all *p* > 0.05)
[36]	Fatigue (measured by Fatigue assessment: created by Piper et al. 1998)		✓			The mean (±SD) fatigue scores were 2.98 ± 1.96 and 3.99 ± 1.64 for the control and the trial group, respectively, and these figures were statistically significant (*p* = 0.036).
[37]	reduction symptom burden (measured by Symptom Distress Scale (SDS) Area Under Curve (AUC) calculation)			✓		There was no difference between groups in mean SDS AUC, adjusted for baseline (MA mean ¼ 25.5, SD ¼ 8.3; MR mean ¼ 25.3, SD ¼ 8.5; *p* ¼ = 0.505).
[38]	Fatigue (measured by Multidimensional Fatigue Symptom InventoryeShort Form (MFSI-SF)).	✓				The Tai Chi group had a lower MFSI-SF total score compared with the control group (53.3 ± 11.8 vs. 59.3 ± 12.2, *p* < 0.05).
[39]	Fatigue/self efficacy (multiple measures used for assessment)	✓				All of the primary contrasts of interest were significant (all *p* < 0.05)

**Table 5 ijerph-19-00536-t005:** Socio-Demographic Variables by Study.

Study/Sociodemographic Variable	Age	Sex	Level of Education	Smoking Status	Marital/Living Arrangement
[29]	mean age 64.165	53% female 47% male	mean of 10.64 years	not specified	83% married 17% not married
[30]	mean age 63.575	56% female 44% male	mean of 10.71 years	not specified	82% married 18% not married
[31]	mean age 63.55	55% male 45% female	53% completed high school as a minimum	18.5% never 54.5% former 27% current	81.5% do not live alone
[33]	mean age 56.2	31.5% male 68.5% female	65.5% high school and above	91.5% never 7% former 1.5% current	not specified
[34]	≤60 36.5% 61–70 31% ≥70 32.5%	60% male 40% female	not specified	not specified	not specified
[35]	mean age 63.05 53% ≤65	60.3% male 39.7% female	not specified	not specified	not specified
[36]	mean age 56.1 maximum age 65	68% male 32% female	not specified	not specified	not specified
[37]	mean age 60.6	49% male 51% female	82% high school and above	not specified	not specified
[38]	≤60 56% >60 44%	75% male 25% female	35% high school or above	53% never 47% smoker	87% partnered
[39]	mean age 67.58	40% male 60% female	74% high school or above	not specified	77% married

## Data Availability

Data is contained within the article.

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
