# Peer review of "Lung Cancer and Self-Management Interventions: A Systematic Review of Randomised Controlled Trials"

_ijerph, 2022, doi:10.3390/ijerph19010536_

Round 1

Reviewer 1 Report

Thank you for giving me this opportunity to review this article. This systematic review concluded that self-management interventions improve outcomes among lung cancer patients and consideration to patient characteristics may predict self-management effect. In my opinion, this article may be improved by the following requests.

Major issues:

  1. Self-management interventions is the main part of this review. However, the definition of self-management interventions was not well described enough for readers to compare them to standard care. Please describe the definition of self-management interventions more specifically in the Introduction and Methods.
  2. Details in both the intervention and control is equally important in the review. Please also clarify the details in the intervention and control in Table S1. I strongly suggest to move Table S1 to Table 1. This table is the main part of this article.
  3. The outcome needs to be specified in the Methods part and also be included in the Table 1 (especially the actual value of the results of each study). It is not suggested to provide only the P values without the actual values and their corresponding confidence intervals.  
  4. Please explain why the authors did not conduct a meta-analysis in the discussion. 
  5. Could the authors elaborate the novelty of this article more? The readers may not understand what this study add to the current literature. If there were already systematic review and meta-analysis that has similar conclusion, then why do we need this article?
  6. Please compare the evidence of this review to the current lung cancer guideline about self-management interventions.
  7. In the discussion, please discuss potential confounding factors that may influence the results in the included studies.
  8. The appraisal of each RCT was very important in this article. Please use RoB 2.0 tool and provide the results as Table 2. Biases in each study may also be discussed in the Discussion.
  9. Is there any limitations in this study? Please provide the limitations in the Discussion.
  10. What is the role of self-management in lung cancer when there are already various treatments in lung cancer guidelines? The role of self-management in different stages of lung cancer may also be explained more. The application of GRADE may also be useful.

Minor issues:

In Figure 1, I believed that the studies that were excluded not because of some "wrong" issues but only because they did not fulfill the inclusion criteria. The authors need to define the inclusion criteria more specifically. Moreover, the left column was quite confusing (8 RCTs to 2 RCTs to 10 RCTs?). Please use the latest version of PRISMA 2020 to improve Figure 1.

Author Response

Dear Experts,

Thank you so much for taking the time to read this manuscript and provide commentary. I appreciate the feedback immensely, and I have addressed each point to the best of my ability below.

Kind Regards

Rachel Rowntree

Reviewer 2 Report

Dear Dr. Rowntree and Dr.Hosseinzadeh,

Thanks for your high-quality manuscript on lung cancer and self-management interventions. I will accept it after a minor revision.

  1. You mentioned Supplementary Tables S1, S2 on Line 83. However, I cannot find it in this manuscript version, and the link is not working. Please submit those parts of the data for review.
  2. I also reviewed your data based on PRISMA 2009 Checklist(Line 64). For the eligibility criteria under the methods part(PRISMA 2009 Checklist), please list report characteristics(Line 69). Please list the years of the publication you considered and clarify whether you included the articles based on other languages and the publication status of those articles.
  3. For Table 2(Line 140), please list the study size for each article.

Congratulations!

Author Response

(The authors gave the same response as above.)

Reviewer 3 Report

Dear Editor,

It was a pleasure to review the review article titled "Lung Cancer and Self-Management Interventions: A Systematic Review of Randomised Controlled Trials" submitted for publication by Rachel Rowntree and Hassan Hosseinzadeh.

Although the topic is interesting, I believe the manuscript needs intensive work, and should be completely polished for language.

Below my specific comments:

Abstract:

  1. Needs polishing, e.g.:
  2. Avoid starting sentences with numbers, e.g., 587 studies...
  3. Six studies met their primary endpoint (endpoint should be plural)

Introduction

  1. I would delete the first sentence. This review is about lung cancer treatment and not cancer and aging.
  2. Add references to all important claims, particularly the third paragraph
  3. Last line: RCT should have an s.

Materials and Methods

  1. Generally past tense
  2. Table 1 is a bit confusing: exclusion for patients looks like a design issue. In addition, it needs a foot note.
  3. Other common exclusion criteria such as duplicate reports are missing

Results

  1. Not sure what the section before 3.1 is about. Should this have a subtitle?
  2. I think this section needs improved organization. This part is hard to follow. Sometimes, descriptions are too shallow, e.g., 3.4

Discussion

  1. Start the section with a clear statement of your findings and maybe after you reinstate the aim.
  2. All study limitations should be mentioned
  3. The conclusion should be concise. It has a lot of background information

Author Response

(The authors gave the same response as above.)

Round 2

Reviewer 3 Report

Dear Editor,

The authors have addressed the majority of my concerns. However, a few points were not addressed properly:

1. "Methods should be past tense in general". There are still present and future tenses there. I also believe the manuscript still needs polishing

2. “Not sure what the section before 3.1 is about. Should this have a subtitle?” in Results remains. There is still a chunk of text before 3.1 with no subtitle.

Author Response

Thank you for your recent email. Please find attached the revised manuscript which addresses the minor revisions set out by reviewer three. All changes (additions and deletions) are marked in red. 
